# Faecal Viral Excretion and Gastrointestinal Co-Infection Do Not Explain Digestive Presentation in COVID-19 Patients

**DOI:** 10.3390/microorganisms11071780

**Published:** 2023-07-09

**Authors:** Inès Rezzoug, Benoit Visseaux, Mélanie Bertine, Marion Parisey, Christine Bonnal, Etienne Ruppe, Diane Descamps, Jean François Timsit, Yazdan Yazdanpanah, Laurence Armand-Lefevre, Sandrine Houze, Nicolas Argy

**Affiliations:** 1Laboratoire de Parasitologie-Mycologie, Hôpital Bichat-Claude Bernard, Assistance Publique des Hôpitaux de Paris, 75018 Paris, France; ines.rezzoug@aphp.fr (I.R.); christine.bonnal@aphp.fr (C.B.); sandrine.houze@aphp.fr (S.H.); 2Laboratoire de Virologie, Hôpital Bichat-Claude Bernard, Assistance Publique des Hôpitaux de Paris, 75018 Paris, France; benoit.visseaux@aphp.fr (B.V.); melanie.bertine@aphp.fr (M.B.); diane.descamps@aphp.fr (D.D.); 3IAME Unit, INSERM, Faculté de Médecine, Université de Paris Cité, 75018 Paris, France; etienne.ruppe@aphp.fr (E.R.); jean-francois.timsit@aphp.fr (J.F.T.); yazdan.yazdanpanah@aphp.fr (Y.Y.); laurence.armand@aphp.fr (L.A.-L.); 4Service des Maladies Infectieuses et Tropicales, Hôpital Bichat-Claude Bernard, Assistance Publique des Hôpitaux de Paris, 75018 Paris, France; 5Laboratoire de Bactériologie, Hôpital Bichat-Claude Bernard, Assistance Publique des Hôpitaux de Paris, 75018 Paris, France; 6Service de Réanimation Médicale et Infectieuses, Hôpital Bichat-Claude Bernard, Assistance Publique des Hôpitaux de Paris, 75018 Paris, France; 7MERIT UMR 261 Unit, IRD, Faculté de Pharmacie, Université de Paris Cité, 75006 Paris, France

**Keywords:** COVID-19, gastrointestinal, enteric pathogens, viral shedding

## Abstract

The physiopathological mechanisms responsible for digestive symptoms in COVID-19 patients are still unclear. The aim of this study was to determine the influence of faecal viral shedding on digestive symptoms and propose differential diagnoses in order to understand the gastrointestinal clinical spectrum in acute cases of COVID-19. All patients managed between March and May 2020, from whom stool samples were collected for microbiological investigations, were included. Microbiological analysis consisted of syndromic PCR screening and microscopic parasitological examination supplemented with microsporidia and multiplex protozoa PCR. SARS-CoV-2 infection was diagnosed via viral detection in respiratory and frozen stool samples, completed via serological test when necessary. Epidemiological, clinical, radiological, and biological data and clinical courses were compared according to COVID-19 status and faecal SARS-CoV-2 shedding and enteric co-infection status. The sample included 50 COVID+ and 67 COVID− patients. Faecal viral shedding was detected in 50% of stool samples and was associated with a higher viral load in the upper respiratory tract. Detected enteric pathogens were not different between subjects with different COVID-19 statuses or faecal SARS-CoV-2 shedding and had no impact on the clinical course for COVID-19 patients. The connection between SARS-CoV-2 shedding and enteric pathogen co-infection involvement in gastrointestinal presentation and clinical course is still unclear, suggesting other processes are involved in digestive disorders in COVID-19 patients.

## 1. Introduction

A new coronavirus, called SARS-CoV-2, emerged in December 2019 from Wuhan, Hubei Province, China [1] and rapidly became a pandemic health care problem; currently, 192 countries worldwide are affected, including, as of February 2023, 673 million confirmed cases and more than 6 million deaths.

This enveloped RNA virus, transmitted by droplets, aerosols, and direct contact, belongs to the betacoronavirus genus and is primarily implicated in a SARS-like atypical pneumonia associated with bilateral ground-glass opacity in chest CT scans [2,3,4,5,6]. However, extra-pulmonary manifestations have also been described in COVID-19 patients. Among them, gastrointestinal disorders have been observed in 1 to 79% of infected patients, with various symptoms including anorexia, diarrhoea, nausea, vomiting, and abdominal pain [2,3,4,5,6,7,8,9] sometimes appearing before the respiratory symptoms [8,9,10,11]. 

While the gastrointestinal tropism of SARS-CoV-1 and MERS-CoV is well-described, the mechanisms underlying digestive disorders in SARS-CoV-2 are still under investigation. Similar to SARS-CoV-1, SARS-CoV-2 penetrates alveolar, digestive and hepatic cells via interactions between the viral spike S glycoprotein and the host receptor angiotensin-converting enzyme 2 (ACE2), followed by membrane fusion enhanced by the co-expressed serine proteases TMPRSS2 [12]. Previous studies reported the presence of intracytoplasmic nucleocapsid in digestive biopsies and prolonged viable viral particle shedding in stool samples from COVID-19 patients, suggesting intestinal viral multiplication and a potential risk of faecal–oral transmission of SARS-CoV-2 [13,14,15,16]. However, the relationship between SARS-CoV-2 faecal shedding and digestive disorders is still unclear [9,17]. In addition to the hypothesis of a direct effect of the virus on gastrointestinal presentation, the influence of host-related or infection-related factors has also been suspected [18]. Alterations of the gut microbiome via metabolic and/or inflammatory mechanisms have also been observed in SARS-CoV-2 patients [19,20], and the resultant dysbiosis was thought to facilitate and sustain gastrointestinal involvement via opportunistic pathogen proliferation, intestinal inflammation, and metabolic disorders [18,19,20,21,22]. Although early observations of COVID-19 patient cohorts in China and later in the USA and Europe considered SARS-CoV-2 the main causative agent in these intestinal disorders, few studies included a comprehensive investigation of the differential diagnosis of gastrointestinal disorders described in COVID-19 patients [23,24].

During the first French epidemic wave in the spring of 2020, SARS-CoV-2 faecal shedding, as well as potential gastrointestinal co-infections during the clinical course of COVID-19 patients, were investigated in a French hospital in order to better identify the role of SARS-CoV-2 infection in digestive symptoms and to optimise clinical management.

## 2. Materials and Methods

### 2.1. Ethical Statement

Ethical review and approval were waived for this study due to thatthis single centre study was performed according to current French legislation for the protection of individuals during health care management (https://www.legifrance.gouv.fr/codes/article_lc/LEGIARTI000045629992/ (accessed on 22 April 2022)) and used existing medical data. No informed consent was required for the study other than the lack of refusal by the patient to participate in medical research. Information about the possibility of using patients’ medical data, unless they express their opposition, is present in their medical reports. In cases where patients refused permission to use their data, those patients were not included.

### 2.2. Study Population

This prospective study was conducted from 1 March 2020 to 3 May 2020 during the first wave of the SARS-CoV-2 pandemic in France. All patients receiving treatment at Bichat-Claude Bernard Hospital who provided stool samples for microbiological investigations were included, regardless of the reason for care. For patients with suspected COVID-19, SARS-CoV-2 molecular diagnostic tests were performed on respiratory samples on admission, with additional SARS-CoV-2 molecular detection in stool samples. In cases where the clinical picture was strongly suggestive of COVID-19 (typical clinical presentation and history and radiographic detection of ground-glass opacities) but respiratory samples were negative, SARS-CoV-2 serology in the month following the acute infectious episode was performed to confirm COVID-19 (COVID+ group). Patients with SARS-CoV-2 RNA detected in their stool samples were assigned to the excretory group as opposed to patients with undetectable faecal viral shedding (non-excretory group). Patients who presented an atypical clinical presentation and history without typical pulmonary radiography of COVID-19 and/or negative results for SARS-CoV-2 PCR on respiratory samples were retrospectively and prospectively included, according to the above-mentioned criteria, from 1 February 2020 to 3 May 2020, to constitute a negative control group for gastrointestinal co-infections (COVID− group). Nosocomial COVID-19 infection (i.e., proven SARS-CoV-2 infection acquired during hospital stay) and onset of digestive symptoms later than 30 days after hospital care were excluded. For each included patient, epidemiological (age, sex, place of residence, travel history), current treatment (background and anti-infectious treatment), clinical (underlying disease, disease history, type and duration of clinical signs, clinical course and outcomes), radiological (extent of pulmonary ground-glass opacities), and biological data (sampling, microbiological, haematological and biochemical investigations) were collected from medical records.

The aetiology of digestive symptoms was determined by assessing current medical treatment, underlying disease, disease history, type and duration of gastrointestinal symptoms and microbiological results. When possible, the onset and duration of digestive and respiratory symptoms, as well as the interval between the onset of symptoms and the start of medical care, the delay between digestive and respiratory symptoms, length of hospital stay, respiratory and stool sampling (i.e., the time between the onset of clinical symptoms and biological sampling) as well as the delay between digestive and respiratory sampling were extracted from patient records. ICU management was taken into account from at least day 1 of hospitalisation in the unit. The collected underlying diseases were chosen because they had already been considered as risk factors for severe COVID-19 [3,4]. COVID-19 was considered severe if infected patients were managed in the ICU and/or if they died. Fever was defined as a body temperature over 38 °C. Diarrhoea was defined as a modification of stool consistency associated with an increased frequency of bowel movements. Diarrhoea was considered chronic if it persisted for at least 30 days. 

### 2.3. Virological Investigations

The RealStar^®^ SARS-CoV-2 RT-PCR Kit 1.0 assay (Hamburg, Germany), targeting the SARS-CoV-2 N and E genes was used for viral detection. RNA extraction was performed with the MagNA Pure LC 2.0 System (Roche, Basel, Switzerland) according to the manufacturer’s protocol. For each sample, 200 μL of nasopharyngeal (NP) sample or faeces suspension was diluted in 2 mL of lysis buffer and eluted in 50 μL. A total of 10 μL of extracted RNA was used to perform the real-time RT-PCR with an ABI Prism^®^ 7500 SDS (Applied Biosystems, Waltham, MA, USA). A heterologous amplification system (Internal Control), included in the kit, was used to identify possible RT-PCR inhibition. In case of inhibition, NP and faeces samples were retested under the same conditions at a 1:10 dilution in PBS. SARS-CoV-2 anti-N antibody detection was performed using Abbott Architect SARS-CoV-2 immunoglobulin G (IgG) (Abbott, Maidenhead, UK) and expressed as an index (cutoff: 0.49).

### 2.4. Microbiological Investigation on Stool Samples

A syndromic approach using QIAstat-Dx^®^ gastrointestinal Panel (Qiagen^®^, Hilden, Germany), which targets 14 bacteria (*Vibrio vulnificus*, *Vibrio parahaemolyticus*, *Vibrio cholerae*, *Campylobacter* spp. (*C. jejuni*, *C. upsaliensis*, *C. coli*), *Salmonella* spp., *Clostridioides difficile* (tcdA/tcdb), *Yersinia enterocolitica*, enterotoxigenic *Escherichia coli* (ETEC), enteropathogenic *E. coli* (EPEC), enteroaggregative *E. coli* (EAEC), *E. coli* producers of shigatoxins (STEC) (enterohaemorrhagic *E. coli*/serotype O157:H7), enteroinvasive *E. coli* (EIEC)/*Shigella* spp., *Plesiomonas shigelloides*), 4 viruses (human Adenovirus F40/F41, Norovirus (group 1 and 2), Rotavirus A, Astrovirus and Sapovirus (group 1, 2, 4, 5)) and 4 parasites (*Entamoeba histolytica*, *Cryptosporidium* spp., *Giardia intestinalis* and *Cyclospora cayetanensis*) were performed on stool samples according to manufacturer’s instructions. In case of positive results for *C. difficile*, detection of the free toxins A and B was performed by membrane enzyme immunoassay (CDIFF QUIK CHEK COMPLETE^®^, Alère^®^, Waltham, MA, USA). 

Trained microscopists also performed microscopic parasitological examinations on the same stool sample after iodine-stained wet mount, flotation, and biphasic concentration. In addition, stool smears stained by modified Ziehl-Neelsen, Microsporidia PCR (Bio-evolution^®^, Bussy-Saint-Martin, France) and a 4-plex protozoa PCR targeting *E. histolytica*, *Cryptosporidium* spp., *G. intestinalis* and *D. fragilis* (Amplidiag^®^, Mobidiag^®^, Espoo, Finland) were also systematically performed according to the manufacturer’s instructions. DNA was extracted from stool on EZ1^®^ Advanced XL (Qiagen^®^) using the EZ1^®^ DNA tissue kit and the EZ1 bacteria card with PBS suspensions of stool samples pre-treated with proteinase K. 

A diagnosis of gastrointestinal co-infection was recorded if an enteric microorganism, known as a possible pathogen, was detected in a patient who presented digestive symptoms without other evident aetiology combined with a high microorganism burden (positive microscopic examination and/or Ct < 30 on QIAstat-Dx^®^ and/or Ct < 35 on Amplidiag^®^). Otherwise, carriage of the detected microorganisms was considered.

### 2.5. Statistical Analysis

Quantitative variables were represented by the mean and standard deviation or by the median and interquartile range [25th percentile–75th percentile] according to their distribution. Categorical variables were evaluated according to size and frequency. Fisher’s exact test was used to compare categorical variables between groups. Kruskall-Wallis tests were used to compare quantitative variables between groups when appropriate. For all tests, a difference was considered significant when *p* < 0.05. All reported *p* values are two-tailed. Statistical analyses were performed using STATA, version 12 (Stata corp^®^, College station, TX, USA).

## 3. Results

### 3.1. Patient Characteristics

During the study period, 117 patients were included. Among them, 50 patients were managed for SARS-CoV-2 infection (42.7%) and 67 were SARS-CoV-2 non-infected patients (57.3%) managed, during the same period, for their underlying disease (neoplasia, solid organ transplant, chronic cardiovascular or pulmonary disease, HIV) (Table 1). COVID-19 was initially diagnosed in nasopharyngeal swabs (84%) (42/50), in BAL (2%) (1/50), in stool (4%) (2/50) or by serology (10%) (5/50). COVID-19 patients were treated with corticoids (44%) and/or hydroxychloroquine (6%) and/or lopinavir/ritonavir (36%) and/or tocilizumab (6%) and/or anakinra (14%) and antibiotics (74%) to prevent secondary bacterial infection. Twenty percent of COVID-19 patients did not receive any treatment. For the COVID− group, curative treatment consisted of antibiotics (56.7%), anti-tuberculosis therapy (6%), antifungal treatment (3%), antiviral treatment (3%) and/or antiparasitic therapy (1.5%). A total of 54% of patients received no treatment.

Both study groups had similar characteristics, except that COVID-19 patients had a significantly higher body mass index (BMI, *p* < 0.001), a higher incidence of hypertension (*p* = 0.008) and presented acute digestive symptoms for a shorter period (7 [5–15] days versus 16 [5–41] days; *p* = 0.03) associated with general clinical symptoms defined by fever, myalgia, dysgeusia and anosmia (*p* < 0.001) (Table 1). In addition, a significantly lower lymphocyte count and higher SGPT were observed in the COVID+ group compared to COVID− patients (*p* < 0.001). Clinical course and management were comparable in both groups.

### 3.2. Virological Features

Virus excretion in stools was detected in 50% of COVID+ patients. Epidemiological characteristics, clinical-radiological presentation, biological data, disease history and clinical course were similar between patients with positive or negative faeces for SARS-CoV-2 (Table 2). 

Although underlying disease types were also similar between groups, SARS-CoV-2 PCR-positive stools seemed to be observed more frequently in COVID-19 patients with a greater number of underlying diseases than the non-excretory group (*p* = 0.04) (Table 2). Seventy-six percent of patients in the excretory group presented at least one digestive symptom compared to 88% in the non-excretory group (Table 2). A link was observed between faecal viral excretion and high SARS-CoV-2 viral load in respiratory samples (*p* = 0.001) (Figure 1). Any influence of the stool sampling (9 days [4–17] vs. 8 days [2–16]; *p* = 0.94), respiratory sampling (7 days [2–12] vs. 6 days [2.5–12]; *p* = 0.79) and the period between each sample (9 days [4–13] vs. 6 days [2–11]; *p* = 0.34) was monitored between the previously defined excretory and non-excretory groups. 

### 3.3. Differential Diagnosis of Gastrointestinal Disorders

Overall, after reviewing the individual medical charts of the included patients, underlying diseases could be the cause of gastrointestinal symptoms in 30% of patients (20/67) in the COVID− group and 4% (2/50) in the COVID+ group (*p* < 0.01) whereas iatrogenic aetiology was suspected in 20.9% (14/67) in the COVID− group but not in COVID+ patients (*p* < 0.01). Aetiology remained unclear in only 6% of COVID− patients (4/67).

Microbiological investigations detected enteric microorganisms in 38% (19/50) and 40% (27/67) of COVID+ and COVID− stool samples respectively (*p* = 0.8). In COVID+ patients with positive stool, 28 enteric pathogens were detected (*C. difficile*, EAEC, EPEC, STEC, *G. intestinalis*, *D. fragilis*, *Blastocystis* spp., *E. dispar* complex, *E. vermicularis* and sapovirus) whereas 37 pathogens were present in COVID− patients (*Campylobacter* spp., *Salmonella* spp., *C. difficile*, *Y. enterocolitica*, EIEC/*Shigella* spp., EAEC, EPEC, STEC, *G. intestinalis*, *Cryptosporidium* spp., *D. fragilis*, *Blastocystis* spp., *E. hartmanii*, *E. nana*, *E. coli*, *H. nana* and Norovirus groupe 1 and 2) (*p* = 0.9) (Figure 2). Twenty-eight percent of COVID+ and 27% of COVID− stool samples tested positive for potentially commensal microorganisms, and 16% of COVID+ and 19% of COVID− stool samples tested positive for known pathogens (*p* = 0.8) (Figure 2A,B).

Presumed food contamination was suspected in 32% and 37% of COVID+ and COVID− groups (*p* = 0.7) whereas faecal-oral transmission of gastrointestinal microorganisms was suspected in 100% and 93% respectively (*p* = 0.5). The number of stools with enteric pathogens, the total number of detected enteric pathogens, the number of pathogenic, commensal or total microorganisms (bacteria, virus and parasite) detected and the number of each pathogenic, commensal or total detected species were compared between COVID+ and COVID− patient groups (Figure 2) as well as between excretory and non-excretory RNA SARS-CoV-2 patients and no difference was observed in these patient group for the tested variables. Moreover, no influence of the detected gastrointestinal microorganisms was observed on either clinical course or outcomes in COVID-19 patients (Table 3).

## 4. Discussion

SARS-CoV-2 is an emerging respiratory virus that preferentially targets the upper respiratory tract and spreads extensively through aerosol transmission and direct contact [25]. Several observations have also implied SARS-CoV-2 in digestive symptoms potentially explained by gastrointestinal tract tropism as already described for SARS-CoV-1 [12]. However, the inconsistent presence of gastrointestinal symptoms in COVID-19 and the lack of knowledge of the underlying physiopathological mechanisms of gastrointestinal infection by SARS-CoV-2 challenge the idea of direct viral involvement in digestive complications. Differential diagnosis of gastrointestinal symptoms in COVID-19 patients is warranted to better understand the spectrum of this viral infection and to optimise clinical management with targeted therapy [23,24]. The aim of the work described here was to explore the involvement of viral shedding in gastrointestinal symptoms and possible differential diagnoses in COVID-19 patients managed in our hospital centre during the first wave of the pandemic in France.

Despite specific hospital organisation during the first wave of SARS-CoV-2, with hospital care management only for symptomatic patients at-risk of severe COVID-19, our patient population is consistent with previous Asian and US patient cohorts [9]. Gastrointestinal presentation was observed in patients at risk of developing severe COVID-19 (i.e., with higher BMI, high incidence of hypertension, elevated SGPT levels and low lymphocyte count) with a significant mortality of 18% [7,8,9,10,11,26]. The early onset and short duration of digestive signs observed in the COVID+ population during clinical management, associated with systemic symptoms such as fever, myalgia and ageusia/anosmia, is in line with COVID-19 acute digestive episodes. In addition, only COVID− patients with severe clinical conditions were hospitalized during the first wave of the SARS-CoV-2 pandemic, which could explain the more prolonged digestive presentation more frequently resulting from underlying diseases or associated medications.

Faecal viral shedding has no impact on digestive presentation and clinical course in COVID-19 patients as previously observed in Chinese and US cohorts [6,8,9,17,27] challenging the hypothesis of a direct viral cytolytic effect on infected enterocytes. However, faecal SARS-CoV-2 excretion seems to represent a severity marker as stools positive for SARS-CoV-2 RNA were more frequently observed in patients with higher numbers of underlying diseases associated with severe COVID-19, and correlated with a high viral load in upper respiratory samples resulting from lack of control of viral multiplication by innate immunity in the population at risk of severe COVID-19 [18,27,28]. Although a small number of studies have demonstrated the presence of viable viral particles in stool and in enterocytes of COVID-19 patients [13,14,15,16], our results could not exclude passive passage of altered SARS-CoV-2 through the carriage of respiratory mucus from the upper respiratory tract to the gut [15]. The most frequent detection of SARS-CoV-2 in patients reported sputum production is also an argument for protected passive viral passage through swallowing mucus [15,16]. Therefore, faecal viral shedding may more likely reflect a high respiratory viral load as opposed to viral multiplication in the gastrointestinal tract. An indirect role of viral particles in stool was also suspected by inducing intestinal pro-inflammatory IL-18 response leading to altered intestinal microbiota, systemic inflammation, and the onset of a cytokine storm [29]. Our study’s focus was on digestive symptoms compared between small excretory and non-excretory groups which could explain the absence of differences between the variables assessed. Additional data would be necessary to explore these hypotheses further.

Gastrointestinal infectious diseases seem to be the main cause of the observed digestive symptoms in COVID-19 patients. Differential microbiological diagnostic revealed that, among the 38% of SARS-CoV-2 stool samples containing enteric microorganisms, approximately 20% were co-infected with known gastrointestinal pathogens. The incidence of gastrointestinal pathogens was similar between COVID+ and COVID− patients, in contrast with previous studies which reported a higher incidence of *C. difficile* in COVID+ patients [24]. One of the physiopathological hypotheses for enteric symptoms during COVID-19 is an alteration of the intestinal microbiome through dysregulation of the lung-gut axis promoting a dysbiosis, characterised by a reduction in the diversity, an initiation of an inadequate systemic and local immune response and an alteration in the host’s metabolism, which contribute to the maintenance of digestive damage and proliferation of opportunistic intestinal pathogens [18,19,20,21,29,30,31,32]. Similar distributions of enteric commensal and known pathogenic bacteria, viruses, and parasites between COVID and excretory groups were observed, indicating, firstly, that COVID-19 patients are not overexposed to faecal-oral transmission compared to our control group and, secondly, that, SARS-CoV-2 infection and faecal shedding do not specifically promote gastrointestinal pathogen infection. However, we noticed trends concerning the larger panel of gastrointestinal pathogens detected in COVID− patients compared to COVID+ patients (18 different pathogens versus 10 in COVID+ patients) (Figure 2). Loss of protozoan and helminth pathogens coupled with a non-significant increase of *Blastocystis* spp., *D. fragilis*, EAEC, EPEC and *C. difficile* from COVID− to COVID+ patients could be explained by the initiation of an altered gut microbiome associated with intestinal inflammation. The presence of non-pathogenic amoeba (*Blastocystis* spp., *D. fragilis*, *Entamoeba* spp., *Endolimax nana*), helminths, or *Cryptosporidium* spp. in the gut was initially considered beneficial for the host, contributing to the diversity of bacteria microbiota and to an intestinal anti-inflammatory response [33,34,35]. By contrast, EPEC, EAEC, as well as the intestinal proliferation of *Blastocystis* spp. and *D. fragilis*, have been associated with the irritable bowel syndrome (IBS) [36]. Complementary studies would be necessary to confirm a possible IBS-like gastrointestinal presentation during severe COVID-19. Microbiological investigations for differential diagnosis are warranted in cases of gastrointestinal presentation during SARS-CoV-2 infection to propose targeted anti-infectious therapy against frequent community-acquired diarrhoea. However, in our study context, no impact of gastrointestinal microorganisms was observed on the clinical course of COVID-19 patients and on SARS-CoV-2 shedding in stool. The first possible explanation is methodological, as the small size of the study group and the low number of enteric pathogens detected, resulting from incomplete microbiological investigations, did not allow us to identify differences between the groups. The second possible explanation is pathophysiological, with a possible predominant effect of dysbiosis induced by COVID-19 compared with gastrointestinal infection by conventional pathogens. This latter hypothesis would corroborate our previous observations on the lack of impact of faecal viral excretion on the gastrointestinal presentation of COVID-19.

This work was designed as a preliminary prospective single centre proof-of-concept study during the first wave of the COVID-19 pandemic in France to explore the frequency and impact of differential gastrointestinal diagnoses in COVID-19. This design presented some limits. The particular context in which this work was performed during the first wave of the COVID-19 pandemic in France precluded the inclusion of a large numbers of patients, and exhaustive microbiological investigations were not possible. Owing to its single-centre design, microbiological investigations could have biased the panel of enteric pathogens identified. The microbiological approach was based in most cases only on one stool sample, possibly underestimating the diagnosis of parasites. These different factors could have contributed to the lack of differences between patient groups. Moreover, we were not able to recruit non-severe COVID+ and COVID− patients with different underlying diseases which also limited the possibilities of observing the variety in number and types of gastrointestinal pathogens and their impact on clinical course, intestinal microbiome and outcome between patient groups. In order to assess the hypothesis of a systemic impact of dysbiosis during COVID-19, other clinico-biological variables should be chosen to compare patient groups, such as, for example, the lack of data collected regarding co-morbidity factors, the intensity of the digestive presentation or detailed care management, which could also help to explain the absence of difference between groups, emphasising the need for further investigation. Clinical management and anti-infectious therapy were not standardised in this study, leaving it to the discretion of the clinicians, thereby adding heterogeneity to data on clinical course and outcome.

This study highlights the continued ambiguity concerning the nature of viral physiopathological mechanisms in COVID-19 digestive presentation. Whereas faecal RNA viral shedding could simply reflect the passive transfer of high nasopharyngeal viral loads, an indirect viral effect through alteration of gut microbiome seems to be a more plausible explanation considering both the absence of any observable impact of SARS-CoV-2 faecal shedding and gastrointestinal co-infection on the clinical course, and the nature of gastrointestinal pathogens detected in COVID-19 patients associated with an IBS-like presentation. Therefore, the importance of differential microbiological diagnosis in investigating any digestive disorders present in COVID-19 patients appears warranted, since the levels of detection of classic enteric pathogens were similar in non-COVID-19 patients with enteric symptoms.

## Figures and Tables

**Figure 1 microorganisms-11-01780-f001:**
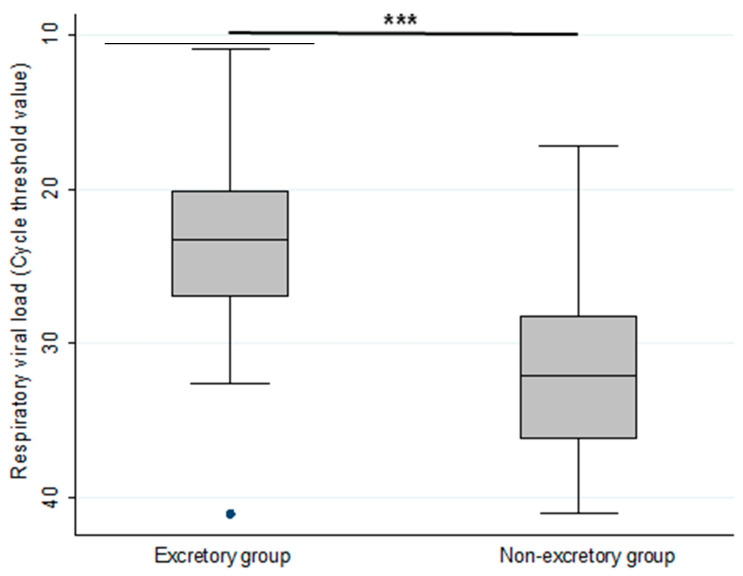
PCR Semi-quantitative respiratory viral load evaluation expressed by Cycle threshold (Ct) value in COVID-19 patients with SARS-CoV-2 RNA positive (excretory group) (*n* = 25) and negative (non-excretory group) faeces. *** *p* value < 0.05.

**Figure 2 microorganisms-11-01780-f002:**
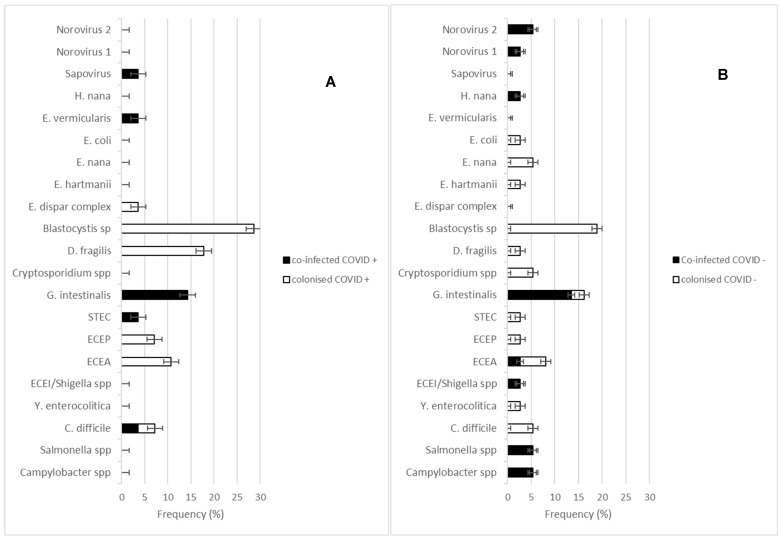
(**A**) The proportion of each gastrointestinal pathogen type responsible for colonisation (colonised) (white scale) or infection (co-infected) (black scale) detected in stool samples of the COVID+ group. (**B**) Proportion of each gastrointestinal pathogen type responsible for colonisation (colonised) (white scale) or infection (co-infected) (black scale) detected in stool samples of COVID6-group.

**Table 1 microorganisms-11-01780-t001:** Epidemiological, clinical and biological data of the included patients receiving hospital care.

Variables	COVID+ (*n* = 50)	COVID− (*n* = 67)	*p* Value
**Epidemiological data**			
Age (years)	60.6 ± 17.8	57.2 ± 20.2	0.34
Sex ratio (M/F)	2.1	1	0.09
Patient’s place of residence			
Out of Paris region	0 (0)	3 (4.5)	0.63
Paris Centre	28 (56)	32 (47.8)	
North of Paris	11 (22)	18 (26.9)	
East of Paris	1 (2)	2 (3)	
West of Paris	8 (16)	12 (17.9)	
South of Paris	2 (3.9)	0 (0)	
European native	19 (38)	37 (55.2)	0.09
Non-European native	31 (62)	30 (44.8)	
Travel history	6 (12)	10 (14.9)	0.79
**Underlying diseases**			
Number of underlying diseases			
0	7 (14)	15 (22.4)	0.53
≤2	30 (60)	35 (52.2)	
>2	13 (26)	17 (25.4)	
Body Mass Index (kg/cm^2^)	25.9 [24.2–31.2]	22.8 [20.5–26.4]	<0.001
Neoplasia	5 (10)	15 (22.4)	0.09
Hypertension	29 (58)	22 (32.8)	0.008
Auto-immune disease	4 (8)	6 (9)	1
HIV	4 (7.8)	5 (7.5)	1
Cardiovascular diseases	8 (16)	17 (25.4)	0.26
Solid Organ Transplant	5 (10)	9 (13.4)	0.78
Diabetes mellitus	13 (26)	16 (23.9)	0.83
Chronic pulmonary disease	4 (8)	9 (13.4)	0.39
None	7 (14)	16 (23.9)	0.28
**Clinical data**			
Number of digestive symptoms			
0	9 (18)	18 (26.9)	0.14
≤2	36 (72)	36 (53.7)	
>2	5 (10)	13 (19.4)	
The onset of digestive symptoms * (days)	−2 [−7–1]	−7 [−29–0]	0.03
Fever	38 (76)	20 (29.9)	<0.001
Diarrhoea	37 (74)	44 (65.7)	0.42
Abdominal pain	11 (22)	21 (31.3)	0.3
Nausea/vomiting	7 (14)	15 (22.4)	0.34
Anorexia	7 (14)	13 (19.4)	0.47
Asthenia	29 (58)	28 (41.8)	0.095
Myalgia	20 (40)	2 (3)	<0.001
Loss of smell/taste	10 (20)	1 (1.5)	<0.001
Duration of digestive symptoms ** (days)	7 [5–15]	16 [5–41]	0.03
Length of hospital stay (days)	14 [8–29]	11 [5–23]	0.1
ICU management	14 (28)	13 (19.4)	0.38
Death	9 (18)	4 (6)	0.07
**Biological data**			
Hb (g/dL)	11.7 [9.9–13.7]	11.4 [9.7–13.1]	0.6
Lymphocyte count (G/L)	1.1 [0.74–1.6]	1.4 [0.85–2.2]	0.03
CRP (mg/L)	40 [8–89]	19 [3–64]	0.1
SGPT (U/L)	38 [22.5–63]	25.5 [17–47]	0.01

Quantitative variables are represented by mean ± standard deviation or as median [1st quartile–3rd quartile]. Categorial variables are represented as *n* (%). Significant values *p* < 0.05. Hb: haemoglobin (normal values: 12.9 g/dL–16.7 g/dL); SGPT: serum-glutamyl-pyruvate-transaminase (normal values: 14 U/L–59 U/L); CRP: C-reactive protein (normal values < 5 mg/L); Lymphocyte count normal values: 1.24 G/L–3.56 G/L. * Interval between onset of digestive symptoms and medical care. ** Interval between onset and end of digestive symptoms.

**Table 2 microorganisms-11-01780-t002:** Epidemiological, clinical and biological data of the included COVID-19 infected patients.

Variables	Excretory (*n* = 25)	Non Excretory (*n* = 25)	*p* Value
**Epidemiological data**			
Age (years)	61.6 ± 17.6	59.6 ± 18.3	0.76
Sex ratio (M/F)	1.3	4	0.13
Non-European native	16 (64)	15 (60)	1
European native	9 (36)	10 (40)	
**Underlying diseases**			
Number of underlying diseases			
0	4 (16)	3 (12)	0.04
≤2	11 (44)	19 (76)	
>2	10 (40)	3 (12)	
BMI (kg/cm^2^)	25.6 [23.5–31.2]	27.3 [25–29.9]	0.39
Neoplasia	3 (12)	2 (8)	1
High Blood pressure	17 (68)	12 (48)	0.25
Auto-immune disease	2 (8)	2 (8)	1
HIV	3 (12)	1 (4)	0.61
Cardiovascular diseases	5 (20)	3 (12)	0.7
Solid Organ Transplant	4 (16)	1 (4)	0.35
Diabetes mellitus	7 (28)	6 (24)	1
Pulmonary background	1 (4)	3 (12)	0.61
**Clinical data**			
Number of digestive symptoms			
0	6 (24)	3 (12)	0.28
≤2	18 (72)	18 (72)	
>2	1 (4)	4 (16)	
Onset of digestive symptoms * (days)	0 [−8–1]	−5 [−7–0]	0.44
Interval between digestive and respiratory symptoms (days) **	5 [0–11]	1 [0–10]	0.62
Diarrhoea	19 (76)	18 (72)	1
Abdominal pain	3 (12)	8 (32)	0.17
Nausea/vomiting	2 (8)	5 (20)	0.42
Anorexia	1 (4)	6 (24)	0.098
Asthenia	15 (60)	14 (56)	1
Duration of digestive symptoms *** (days)	10 [5–16]	7 [5–15]	0.67
Onset of respiratory symptoms (days) †	−7 [−11–−3]	−8 [−12.5–−3.5]	0.72
Fever	20 (80)	18 (72)	0.74
PPI users	9 (36)	6 (24)	0.54
Duration of respiratory symptoms (days) ††	18 [10–26]	22.5 [15.5–34.5]	0.11
Length of hospital stay (days)	13 [9–25]	16 [7–32]	0.99
ICU management	7 (28)	7 (28)	1
Death	5 (20)	4 (16)	1
**Biological data**			
Lymphocyte count (G/L)	1.07 [0.52–1.6]	1.15 [0.84–1.5]	0.78
CRP (mg/L)	45 [8–155]	34.5 [9.5–63]	0.67
SGPT (U/L)	35 [20–59]	40 [25–68]	0.28

Quantitative variables are represented by mean ± standard deviation or as median [1st quartile–3rd quartile]. Categorial variables are represented as *n* (%). Significant values *p* < 0.05. SGPT: serum-glutamyl-pyruvate-transaminase (normal values: 14 U/L–59 U/L); CRP: C-reactive protein (normal values < 5 mg/L); Lymphocyte count normal values: 1.24 G/L–3.56 G/L; PPI: Proton Pump Inhibitors. * Number of days between the onset of digestive symptoms and medical care. ** Number of days between the onset of digestive and respiratory symptoms. *** Number of days between onset and end of digestive symptoms. † Number of days between the onset of respiratory symptoms and medical care. †† Number of days between onset and end of respiratory symptoms.

**Table 3 microorganisms-11-01780-t003:** Epidemiological, clinical, radiological data and clinical course of the included COVID-19 infected patients with a gastrointestinal microorganism (gastrointestinal pathogens and/or commensals) (COVID + enteric pathogens) detected during microbiological investigations on stool samples or without gastrointestinal pathogens (absence of enteric microorganisms) (only COVID).

	COVID + Enteric Pathogens (*n* = 19)	Only COVID (*n* = 31)	*p* Value
**Epidemiological data**			
Age (years)	55.9 +/− 19.2	63.5 +/− 16.5	0.16
Sex ratio (M/F)	2.2	2.1	1
Non-European native	12 (63.2)	19 (61.3)	1
European native	7 (36.8)	12 (38.7)	
**Clinical data**			
Number of digestive symptoms			
0	4 (21.1)	5 (16.1)	0.8
≤2	14 (73.7)	22 (71)	
>2	1 (5.3)	4 (12.9)	
Onset of digestive symptoms * (days)	0 [−6–9]	−2.5 [−11–0]	0.11
Interval between digestive and respiratory symptoms (days) **	4.5 [0–13]	3 [0–9]	0.45
Fever	14 (73.7)	24 (77.4)	1
Diarrhoea	12 (63.2)	25 (80.7)	0.2
Abdominal pain	3 (13.8)	8 (25.8)	0.5
Nausea/vomiting	3 (15.8)	4 (12.9)	1
Anorexia	3 (15.8)	4 (12.9)	1
Asthenia	12 (63.2)	17 (54.8)	0.77
Myalgia	7 (36.8)	13 (41.9)	0.77
Loss of smell/taste	4 (21.1)	6 (19.4)	1
Duration of digestive symptoms *** (days)	7 [5–15]	8 [5–16]	0.5
Onset of respiratory symptoms (days) †	−7 [−11–−4]	−8 [−12–−3]	0.71
Cough	13 (68.4)	19 (61.3)	0.76
Dyspnoea	14 (73.7)	21 (67.7)	0.77
Duration of respiratory symptoms (days) ††	21.5 [10–31]	20 [14–29]	0.7
Hospital stay (days)	10 [4–27]	15 [9–32]	0.2
ICU management	5 (26.3)	9 (29)	1
Death	4 (21.1)	5 (16.1)	0.72
**Radiological data. Ground glass opacity**			
(<10%)	3 (18.8)	6 (22.2)	0.92
(10–25%)	3 (18.8)	4 (14.8)	
(25–50%)	6 (37.5)	8 (29.6)	
(>50%)	4 (25)	9 (33.3)	
**Biological data**			
Lymphocyte count (G/L)	1.4 [0.98–1.7]	1.02 [0.52–1.4]	0.09
CRP (mg/L)	52 [8–89]	29 [8–112]	0.6
ALAT (U/L)	38 [18–67]	38 [24–59]	0.91

Quantitative variables are represented by mean ± standard deviation or as median [1st quartile–3rd quartile]. Categorial variables are represented as *n* (%). Significant values *p* < 0.05. SGPT: serum-glutamyl-pyruvate-transaminase (normal values: 14 U/L–59 U/L); CRP: C-reactive protein (normal values < 5 mg/L); Lymphocyte count normal values: 1.24 G/L–3.56 G/L. * Number of days between the onset of digestive symptoms and medical care. ** Number of days between the onset of digestive and respiratory symptoms. *** Number of days between onset and end of digestive symptoms. † Number of days between the onset of respiratory symptoms and medical care. †† Number of days between onset and end of respiratory symptoms.

## Data Availability

Authors confirm to the editorial microorganisms board that all materials and raw data necessary for manuscript reviewing and comprehension will be from the corresponding author when necessary.

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
