# Peer review of "Faecal Viral Excretion and Gastrointestinal Co-Infection Do Not Explain Digestive Presentation in COVID-19 Patients"

_microorganisms, 2023, doi:10.3390/microorganisms11071780_

Round 1
Reviewer 1 Report
It would be interesting to expand the series with a multicenter study.
I am very interested in learning more about the response in immunocompromised patients (for example oncohaematological patients).
Author Response
It would be interesting to expand the series with a multicenter study.
I am very interested in learning more about the response in immunocompromised patients (for example oncohaematological patients).
Responses: We completely agree with the reviewer's comments. The aim of this single-centre exploratory study was to investigate others possibles aetiologies for the digestive presentations observed in COVID-19 infected patients during the 1st wave of the pandemia in France. These preliminary data obviously need to be extended on a larger scale in multicenter studies that will explore the influence of other types of population and underlying pathologies on gastrointestinal presentation during COVID-19.
Reviewer 2 Report
In this study, the authors assessed the pathogens present in COVID+ patients. The authors compared first between COVID+ and COVID- patients. They found that 50% were excretory. Out of 50 COVID19 +, 19 patients were coinfected, and 31 were COVID19 only.
Major point
1) Please specify the pathogens in COVID-19 -ve. What are the difference in microbial population in both COVID19+, COVID19- patients.
2) Table 3: There was no significant difference between COVID only (n=31) and COVID19+ enteric pathogens. How can the authors explain this?
3) Table 2: There is no significant difference between Excretory and non-excretory . How can the authors explain this?
4) What is the conclusion of coinfection of COVID19 with enteric pathogens, if there is no difference in clinical and lab parameters between the both groups.
Moderate language editing
Author Response
1) Please specify the pathogens in COVID-19 -ve. What are the difference in microbial population in both COVID19+, COVID19- patients?
Responses: In according with reviewer’s comments, we have completed our proof with the list of the enteric pathogens detected in COVID19+ and COVID19- patients. Modifications has been performed from the line 252 of the corrected proof as :” Microbiological investigations detected enteric microorganisms in 38% (19/50) and 40% (27/67) of COVID+ and COVID- stool samples respectively (p = 0.8). In COVID+ patients with positive stool, 28 enteric pathogens were detected (C. difficile, EAEC, EPEC, STEC, G. intestinalis, D. fragilis, Blastocystis spp, E. dispar complex, E. vermicularis and sapovirus) whereas 37 pathogens were present in COVID- patients (Campylobacter spp, Salmonella spp, C. difficile, Y. enterocolitica, EIEC/Shigella spp, EAEC, EPEC, STEC, G. intestinalis, Cryptosporidium spp, D. fragilis, Blastocystis spp, E. hartmanii, E. nana, E. coli, H. nana and Norovirus groupe 1 and 2) (Figure 2). Twenty-eight percent of COVID+ and 27% of COVID- stool samples tested positive for potentially commensal microorganisms, and 16% of COVID+ and 19% of COVID- stool samples tested positive for strict pathogens (p = 0.8) (Figure 2A and 2B).
Secondly, in response to the reviewer's questions, we also repeated the tests comparing the populations of intestinal microorganisms detected between the COVID19+ and COVID19- patient groups. There was no significant difference between the total number of pathogens detected, between the types of pathogens detected (bacteria/viruses/parasites), between the species of microorganisms detected and between the proportions of commensal and pathogenic microorganisms between COVID19+ and COVID19- patients. All these datas were presented in the corrected proof from line 274 as:” The number of positive stools for enteric pathogens, the total number of detected enteric pathogens, the number of pathogenic, commensal or total microorganisms (bacteria, virus and parasite) detected and the number of each pathogenic, commensal or total detected species were compared between COVID+ and COVID- patient groups (Figure 2) as well as between excretory and non-excretory RNA SARS-CoV-2 patients and no difference were observed in these patient group for the tested variables. Moreover, no influence of the detected gastrointestinal microorganisms was observed on clinical course and outcomes in COVID-19 patients (Table 3).”
2) Table 3: There was no significant difference between COVID only (n=31) and COVID19+ enteric pathogens. How can the authors explain this?
Responses: Following the reviewer’s comments, we have developed in the discussion the potential explanation of the absence of difference between patient groups. We though that the absence of impact of fecal viral shedding and co-infection with enteric pathogens reveals a potential role of dysbiosis on gastrointestinal presentation during COVID-19. We developed this idea from line 333 in the corrected proof as “Gastrointestinal infectious diseases seem to be the main cause of the observed digestive symptoms in COVID-19 patients. Differential microbiological diagnostic revealed that, among the 38% of SARS-CoV-2 positive stool samples for enteric microorganisms, approximately 20% were co-infected with strict gastrointestinal pathogens. The incidence of gastrointestinal pathogens was similar between COVID+ and COVID- patients contrary to previous studies, which reported a higher incidence of C. difficile in COVID+ patients [24]. One of the physiopathological hypotheses for enteric symptoms during COVID-19 is an alteration of the intestinal microbiome through dysregulation of the lung-gut axis promoting a dysbiosis, characterised by a reduction in the diversity, an initiation of an inadequate systemic and local immune response and an alteration in the host's metabolism, which contribute to the maintenance of digestive damage and intestinal opportunistic pathogens proliferation [18-21, 29-32]. Similar distribution of enteric commensal and strict pathogenic bacteria, virus and parasite between COVID and excretory groups was observed indicating firstly that, COVID-19 patients are not overexposed to faecal-oral transmission compared to our control group and secondary that, SARS-CoV-2 infection and faecal shedding do not specifically promote gastrointestinal pathogen implantation. However, trends could be noticed concerning the larger panel of gastrointestinal pathogens detected in COVID- patients compared to COVID+ patients (18 different pathogens versus 10 in COVID+ patients) (Figure 2). Loss of protozoan and helminths pathogens coupled to a non-significant increase of Blastocystis spp, D. fragilis, EAEC, EPEC and C. difficile from COVID- to COVID+ patients could be explained by the initiation of an altered gut microbiome associated to intestinal inflammation. Presence of non-pathogenic amoeba (Blastocystis spp, D. fragilis, Entamoeba spp, Endolimax nana), helminths or Cryptosporidium spp in the gut was initially considered as beneficial for the host contributing to the diversity of bacteria microbiota and to an intestinal anti-inflammatory response [33-35]. By contrast, EPEC, EAEC as well as the intestinal protozoa Blastocystis spp. and D. fragilis intestinal proliferation were associated with the irritable bowel syndrome (IBS) [36]. Complementary studies would be necessary to confirm a possible IBS-like gastrointestinal presentation during severe COVID-19.”
3) Table 2: There is no significant difference between Excretory and non-excretory . How can the authors explain this?
Responses: we do not fully agree with the reviewer's comment. In fact, in table 2, a difference in the number of host factors associated with the risk of developing severe COVID-19 was observed between the groups of excretory and non-excretory patients. These results are developed from line 220 of the corrected manuscript as follows :”As Although underlying disease types were also similar between groups, SARS-CoV-2 PCR positive stools appeared to be observed more frequently in COVID-19 patients with a greater number of underlying diseases than non-excretion groups (p = 0.04) (Table 2). Seventy-six percent of patients in the excretory group presented at least one digestive symptom compared to 88% in the non-excretory group (Table 2). An association was observed between faecal viral excretion and high SARS-CoV-2 viral load in respiratory samples (p = 0.001) (Figure 1). Any influence of the stool sampling (9 days [4 – 17] vs. 8 days [2 – 16]; p = 0.94), respiratory sampling (7 days [2 – 12] vs. 6 days [2.5 – 12]; p = 0.79) and period between each sample (9 days [4 – 13] vs. 6 days [2 – 11]; p = 0.34) was monitored between the previously defined excretory and non-excretory groups. “
Modifications were then performed in the corrected proof from line 322 to better discussed these results and try to explain the absence of others difference between our groups: “Faecal viral shedding has no impact on digestive presentation and clinical course in COVID-19 patients as previously observed in Chinese and US cohorts [6,8,9,17,27] challenging hypothesis of a direct viral cytolytic effect on infected enterocytes. However, faecal SARS-CoV-2 excretion seems to represent a severity marker as stools positive for SARS-CoV-2 RNA were more frequently observed in patients with higher numbers of underlying diseases associated to severe COVID-19 and correlated to a high viral load in upper respiratory samples resulting of uncontrolled viral multiplication by innate immunity in at-risk population of severe COVID-19 [18, 27, 28]. Although a small number of studies have demonstrated the presence of viable viral particles in stool and in enterocytes of COVID-19 patients [13–16], our results could not exclude passive passage of altered SARS-CoV-2 through carriage of respiratory mucus from the upper respiratory tract to the gut [15]. The most frequent detection of SARS-CoV-2 in patients reported sputum production is also an argument for protected viral passive passage through mucus swallowing [15,16]. Hence, faecal viral shedding could more likely reflect high respiratory viral load as opposed to viral multiplication in the gastrointestinal tract. Indirect role of viral particles in stool was also suspected by inducing intestinal pro-inflammatory IL-18 response leading to altered intestinal microbiota, systemic inflammation to the onset of a cytokine storm [29]. In our work, focus has been performed on digestive symptoms between a small excretory and non-excretory group which could explain the absence of difference on the chosen variables. Additional data will be needed to explore these hypotheses.”
4) What is the conclusion of coinfection of COVID19 with enteric pathogens, if there is no difference in clinical and lab parameters between the both groups.
Responses: Conclusions of the manuscript was modified in response to reviewer’s remarks as from line 406: “This study highlights the continued ambiguity concerning the nature of viral physiopathological mechanisms in COVID-19 digestive presentation. Whereas faecal RNA viral shedding could simply reflect the passive transfer of high nasopharyngeal viral loads, indirect viral effect through alteration of gut microbiome seems to be a more plausible explanation according to the absence of SARS-CoV-2 faecal shedding impact and gastrointestinal co-infection in clinical course and in front of the nature of gastrointestinal panels detected in COVID-19 patients associated to IBS-like presentation. Therefore, the importance of differential microbiological diagnosis in investigating any digestive disorders present in COVID-19 patients appears warranted, since the levels of detection of classic enteric pathogens were similar in non-COVID-19 patients with enteric symptoms.
Reviewer 3 Report
The manuscript by Inès Rezzoug et al. investigated the effect of faecal viral excretion and gastrointestinal co-infection on the digestive presentation in COVID-19 patients. The study is of interest to the readers and I have the following suggestions and comments:
1, the authors should revise Figure 1. I can hardly read the lebelings.
2, In figure 2, I did not see the error bars. Why? This must be explained.
3, The authors must further discuss the limitations of the current study, This study is a preliminary proof-of-concept study. So many factors could affect the outcomes.
Author Response
1, the authors should revise Figure 1. I can hardly read the lebelings.
Responses: As suggested by the reviewer, we have entirely revised the figure 1 with the objective to make it easier to read and understand. We hope that the modifications performed will be in line with the reviewer’s expectations.
2, In figure 2, I did not see the error bars. Why? This must be explained.
Responses: We apologize to the reviewer for this oversight. The error bars have been added to the revised figure 2.
3, The authors must further discuss the limitations of the current study, This study is a preliminary proof-of-concept study. So many factors could affect the outcomes.
Responses: In accordance with the reviewer’s suggestion, the proof was modified from line 380 as “This work was designed as a preliminary prospective monocentric proof-of-concept study during the first wave of the covid-19 pandemic in France to explore the frequency and impact of differential gastrointestinal diagnoses in COVID-19. This design raised some limits. The particular context in which this work was performed during the first wave of COVID-19 pandemia in France prevented us to include a large size of studied groups as well as performed exhaustive microbiological investigation. Owing to its monocentric design, microbiological investigation could have biased the panel of enteric pathogens identified. The microbiological approach was based in most cases only on one stool sample, possibly underestimating the parasitic diagnosis. All these reasons could contribute to prevented us from observing differences between patient groups. Moreover, we were not able to recruit non-severe COVID+ and COVID- patients with different underlying diseases which also limits the possibilities of observing the variety of the number and types of gastrointestinal pathogens and their impact on clinical course, intestinal microbiome and outcome between patient groups. In the hypothesis of a systemic impact of dysbiosis during COVID-19, others clinic-biological variables should be chosen to compare patient group as for example the lack of data collected regarding co-morbidity factors, intensity of the digestive presentation or detailed care management, which could explain the absence of difference between groups and need to be secondary investigated. Clinical management and anti-infectious therapy were not standardised in this study, leaving it to the discretion of the clinicians and adding heterogeneity to clinical course and outcome data.
Reviewer 4 Report
Rezzoug et al. attempted to mechanistically understand the physiopathology behind the digestive symptoms during COVID-19 infection. In particular, they aimed to investigate the role of fecal viral shedding in influencing digestive symptoms in COVID-19 patients. To achieve this, they leveraged a cohort of 50 COVID+ and 67 COVID- patients recruited at Bichat-Claude Bernard Hospital. They collected their fecal samples and used PCR screening, microscopic examination, and serological tests to diagnose SARS-CoV-2 infection and detect other enteric pathogens. They did not observe a difference concerning the number and species of commensal or pathogenic parasites and bacteria between COVID+ and COVID- patient groups. I am quite open to looking at a revised version if the authors could address some major and minor issues in a satisfactory fashion, which we describe in more detail below.
Major issues:
1. I think the introduction and discussion lack a more detailed discussion of why and how the SARS-CoV-2 virus impacts the infection of other gut pathogens. Please do a summary of previous papers that investigated the potential influence of viruses and cite those papers. For instance, does viral infection change host factors (Logan Miller et al., Viral Immunology 2022)? Or does viral infection change the gut microbiome (Shanlin Ke et al., Nature Communications 2022) then the gut microbiome changes the metabolism (Tong Wang et al., PloS Computational Biology 2019)?
2. Figure 1: The resolution of this figure is not very high. Is it possible to increase the resolution? In addition, the horizontal line below the “***” is left-shifted. Please make sure that it is well-positioned. Also, please write the full name and meaning of “Ct value” in the figure caption so readers can understand it.
3. Figure 2: I don’t quite understand this figure. What is the meaning of “positif”? Also, the color of “COVID+” is highly similar to the color of “colonised COVID+”, making them hard to be differentiated.
Minor comments:
1. Lines 42-45: Please point out the explicit date to which those data were recorded. The number of cases and death changes over time.
2. Line 228: “Overall, after review of…” -> “Overall, after reviewing…”
No major issue detected.
Author Response
Major issues:
- I think the introduction and discussion lack a more detailed discussion of why and how the SARS-CoV-2 virus impacts the infection of other gut pathogens. Please do a summary of previous papers that investigated the potential influence of viruses and cite those papers. For instance, does viral infection change host factors (Logan Miller et al., Viral Immunology 2022)? Or does viral infection change the gut microbiome (Shanlin Ke et al., Nature Communications 2022) then the gut microbiome changes the metabolism (Tong Wang et al., PloS Computational Biology 2019)?
Responses: we would like to thank the reviewers for this pertinent comment. In accordance with these suggestions, we have completed the bibliography for this article and, consequently, we have also modified the introduction and the discussion. Thus, we have introduced the possible influence of intestinal dysbiosis on the digestive tract during COVID-19 and on the proliferation of other gastrointestinal pathogens. Thus, the modifications are present from line 63 of the corrected document such a:” In addition to the hypothesis of a direct effect of the virus on gastrointestinal presentation, the influence of host-related or infection-related factors has also been suspected [18]. Alteration of the gut microbiome through metabolic and/or inflammatory mechanisms was also described during SARS-CoV-2 infection [19, 20], and the resultant dysbiosis was though to facilitate and sustain gastrointestinal involvement through opportunistic pathogens proliferation, intestinal inflammation and metabolic disorders [18-22]. Although early observations on COVID-19 patient cohorts in China and later in the USA and Europe considered SARS-CoV-2 as the main causative agent in the intestinal disorder, few studies included a comprehensive investigation of the differential diagnosis of gastrointestinal disorders described in COVID-19 patients [23,24].” and from the line 333 in the discussion: “Gastrointestinal infectious diseases seem to be the main cause of the observed digestive symptoms in COVID-19 patients. Differential microbiological diagnostic revealed that, among the 38% of SARS-CoV-2 positive stool samples for enteric microorganisms, approximately 20% were co-infected with strict gastrointestinal pathogens. The incidence of gastrointestinal pathogens was similar between COVID+ and COVID- patients contrary to previous studies, which reported a higher incidence of C. difficile in COVID+ patients [24]. One of the physiopathological hypotheses for enteric symptoms during COVID-19 is an alteration of the intestinal microbiome through dysregulation of the lung-gut axis promoting a dysbiosis, characterised by a reduction in the diversity, an initiation of an inadequate systemic and local immune response and an alteration in the host's metabolism, which contribute to the maintenance of digestive damage and intestinal opportunistic pathogens proliferation [18-21, 29-32]. Similar distribution of enteric commensal and strict pathogenic bacteria, virus and parasite between COVID and excretory groups was observed indicating firstly that, COVID-19 patients are not overexposed to faecal-oral transmission compared to our control group and secondary that, SARS-CoV-2 infection and faecal shedding do not specifically promote gastrointestinal pathogen implantation. However, trends could be noticed concerning the larger panel of gastrointestinal pathogens detected in COVID- patients compared to COVID+ patients (18 different pathogens versus 10 in COVID+ patients) (Figure 2). , Loss of protozoan and helminths pathogens coupled to a non-significant increase of Blastocystis spp, D. fragilis, EAEC, EPEC and C. difficile from COVID- to COVID+ patients could be explained by the initiation of an altered gut microbiome associated to intestinal inflammation. Presence of non-pathogenic amoeba (Blastocystis spp, D. fragilis, Entamoeba spp, Endolimax nana), helminths or Cryptosporidium spp in the gut was initially considered as beneficial for the host contributing to the diversity of bacteria microbiota and to an intestinal anti-inflammatory response [33-35]. By contrast, EPEC, EAEC as well as the intestinal protozoa Blastocystis spp. and D. fragilis intestinal proliferation were associated with the irritable bowel syndrome (IBS) [36]. Complementary studies would be necessary to confirm a possible IBS-like gastrointestinal presentation during severe COVID-19.Microbiological investigations for differential diagnosis are warranted in cases of gastrointestinal presentation during SARS-CoV-2 infection to propose targeted anti-infectious therapy against frequent community-acquired diarrhoea. However, in our study context, no impact of gastrointestinal microorganisms was observed on clinical course of COVID-19 patients and on SARS-CoV-2 stool shedding. The first possible explanation is methodological, as the small size of the study group and the low number of enteric pathogens detected, resulting from incomplete microbiological investigations, did not allow us to identify differences between the groups. The second possible explanation is pathophysiological, with a possible predominant effect of dysbiosis induced by COVID-19 compared with gastrointestinal infection by conventional pathogens. This latter hypothesis would corroborate our previous observations on the lack of impact of faecal viral excretion on the gastrointestinal presentation of COVID-19.
- Figure 1: The resolution of this figure is not very high. Is it possible to increase the resolution? In addition, the horizontal line below the “***” is left-shifted. Please make sure that it is well-positioned. Also, please write the full name and meaning of “Ct value” in the figure caption so readers can understand it.
Responses: In accordance with the reviewer’s remarks, we have modified the figure 1 and we hope that the resolution of this figure will be increased. The horizontal line below the “***” was refocused in this new figure version and so well-positioned. “Ct value” name was changed by the entire name “Cycle Threshold value” and its meaning developed in the figure caption as: “Figure 1. PCR Semi-quantitative respiratory viral load evaluation expressed by Cycle threshold (Ct) value in COVID-19 patients with SARS-CoV-2 RNA positive (excretory group) (n = 25) and negative (non-excretory group) faeces.”
- Figure 2: I don’t quite understand this figure. What is the meaning of “positif”? Also, the color of “COVID+” is highly similar to the color of “colonised COVID+”, making them hard to be differentiated.
Responses: We totally agree with the remarks of the reviewer. To better clarify the data exposed, we have reworked the figure and splitted it in a figure 2A and 2B with distinct colors between co infected and colonized patients. We hope this figure 2 should be more comprehensive.
Minor comments:
- Lines 42-45: Please point out the explicit date to which those data were recorded. The number of cases and death changes over time.
Responses: We are totally agreeing with the reviewer. We have by consequence insert the month and year corresponding to the last update of worldwide COVID-19 epidemiology we used for this manuscript as:” A new coronavirus, called SARS-CoV-2, has emerged since December 2019 from Wuhan, Hubei Province, China [1] to rapidly become a pandemic health care problem with currently 192 countries affected worldwide including to date, in February 2023, 673 million confirmed cases and more than 6 million deaths.”
- Line 228: “Overall, after review of…” -> “Overall, after reviewing…”
Responses: The changes have been performed in accordance to the reviewer’s suggestions.
Round 2
Reviewer 2 Report
No further comments
Moderate language editing
Reviewer 3 Report
The authors have revised the manuscript accordingly. It can be considered for publication.
Reviewer 4 Report
The authors answered my questions. I don’t have any more questions.
I didn't detect major grammatical errors.